# Phase stabilization by electronic entropy in plutonium

N. Harrison[1], J.B. Betts[1], M.R. Wartenbe[1], F.F. Balakirev [1], S. Richmond [2], M. Jaime[1] & P.H. Tobash[2]

Plutonium metal undergoes an anomalously large 25% collapse in volume from its largest volume $\delta$ phase ($\delta$-Pu) to its low temperature $\alpha$ phase, yet the underlying thermodynamic mechanism has largely remained a mystery. Here we use magnetostriction measurements to isolate a previously hidden yet substantial electronic contribution to the entropy of $\delta$-Pu, which we show to be crucial for the stabilization of this phase. The entropy originates from two competing instabilities of the 5$f$-electron shell, which we show to drive the volume of Pu in opposing directions, depending on the temperature and volume. Using calorimetry measurements, we establish a robust thermodynamic connection between the two excitation energies, the atomic volume, and the previously reported excess entropy of $\delta$-Pu at elevated temperatures.

[1] Los Alamos National Laboratory, Los Alamos, Mail Stop E536, Los Alamos, NM 87545, USA. [2] Los Alamos National Laboratory, Los Alamos, Mail Stop E574, Los Alamos, NM 87545, USA. Correspondence and requests for materials should be addressed to N.H. (email: nharrison@lanl.gov)

Located at the discontinuity in atomic volume between light and heavy actinides, elemental plutonium (Pu) has an unusually rich phase diagram that includes seven distinct solid state phases and an anomalously large 25% collapse in volume from its $\delta$ phase to its low temperature $\alpha$ phase via a series of structural transitions[1–3]. Despite considerable advances in our understanding of the interplay between the atomic volume and strong electronic correlations within each of the various structural phases of Pu and other actinides[4–15], the thermodynamic mechanism responsible for driving the volume collapse has continued to remain a mystery. Clues to the emergence of additional degrees of freedom at elevated temperatures are provided by calorimetry experiments[16–19], which find the high temperature heat capacity of Pu and neighboring actinides to significantly exceed the Dulong-Petit value of conventional solids[20]. Although electronic degrees of freedom and anharmonic phonons are expected to contribute to the entropy in Pu[6,7], thermodynamic experiments have thus far been unable to isolate a contribution that is sufficiently large to account for the observed stabilization of the $\delta$ phase over the $\alpha$ phase at elevated temperatures[18,19].

Owing to the direct coupling of a magnetic field to magnetic moments, magnetostriction measurements provide a powerful method for isolating the electronic contribution to the lattice thermodynamics[21]. Although the magnetostriction is vanishingly small in conventional non-magnetic metals, it has been shown to become anomalously large in the vicinity of an $f$-electron shell instability[22–26], whereby a thermodynamically accessible difference in energy exists between electronic configurations of the $f$-electron atomic shell comprising different numbers of $f$-electrons. Because magnetic fields do not couple directly to phonons, magnetic field-dependent changes in the phonon contribution occur only in response to a change in the volume that is driven electronically, causing such changes to be a weak higher order effect.

Here we show magnetostriction measurements to uncover the crucial role played by a large electronic entropy originating from the unstable $f$-electron shells[22] of Pu in stabilizing the $\delta$ phase against volume collapse. We find that in contrast to valence fluctuating rare earths, which typically have a single $f$-electron shell instability whose excitations drive the volume in a single direction in temperature and magnetic field[22,27,28], Ga-stabilized $\delta$-Pu exhibits two such instabilities whose excitations drive the volume in opposite directions while producing an abundance of electronic entropy at elevated temperatures. The two instabilities imply a near degeneracy between several different configurations of the 5$f$ atomic shell[29–33], giving rise to a considerably richer behavior than found in rare earth metals. We use heat capacity measurements to establish a robust thermodynamic connection between the two excitation energies, the atomic volume, and the previously reported excess entropy of $\delta$-Pu at elevated temperatures[17–19]. Virtual valence fluctuations[5,8,34], by contrast, in which fluctuations between nearly degenerate $f$-electron shell configurations persist down to zero temperature, due for example to mixing, appear to have little impact on the thermodynamic stabilization of the $\delta$ phase. This is in spite of virtual valence fluctuations having previously been suggested to explain some of the low temperature properties of Ga-stabilized $\delta$-Pu[17,35].

## Results

**Magnetostriction measurements**. We are able to infer the existence of two instabilities in Pu by way of magnetostriction measurements owing to Ga substitution affording the stabilization of the $\delta$ phase over a broad span in temperatures and over a range of different volumes[36,37]. In pure Pu, by contrast, $\delta$-Pu is stable only over a narrow range of high temperatures—collapsing

into significantly lower volume structures upon reducing the temperature. We perform magnetostriction measurements on Ga-stabilized $\delta$-plutonium using an optical fiber Bragg grating technique[38], which we have adapted for use on encapsulated radiologically toxic materials (see Methods).

Figure 1a, b show measurements of the longitudinal magnetostriction (dilation and contraction along the direction of the magnetic field) of Ga-stabilized polycrystalline plutonium samples of composition $\delta$-Pu$_{1-x}$Ga$_x$ with $x$ = 2% and 7%. We find that the magnitude of the electronically driven quadratic-in-magnetic field coefficient of the magnetostriction of $\delta$-Pu (see Fig. 1c, and Supplementary Fig. 1) falls within the range of values observed in materials with unstable $f$-electron shells[22–26]. However, rather than exhibiting a steep upturn at low temperatures[22,23], as expected for a dominant role played by virtual (or zero point) fluctuations involving a low lying magnetic configuration[5,8,34,39], the magnetostriction of $\delta$-Pu is observed to vanish at low temperatures. Its behavior closely resembles that of a scenario in which the $f$-electrons condense into a non-magnetic atomic shell configuration[26,40], revealing the electronic excitations to states with different magnetic configurations to be of a predominantly thermally activated nature.

When excitations to different electronic configurations occur in $f$-electron systems (e.g. from $E_0$ to $E_1$ in Fig. 2a, b), the excited configuration usually has a different number of $f$-electrons confined to the atomic core, causing it to have a different equilibrium atomic volume ($V_1$) and magnetic moment (see Methods)[22,28]. The initial positive increase of the magnetostriction with temperature indicates that the dominant thermal excitations occur between a non-magnetic configuration and a different configuration with both a larger equilibrium atomic size and a larger magnetic moment, as is the case in the majority of $f$-electron systems (see e.g. the illustrated case of Ce in Fig. 2a)[22,23,25,26]. However, rather than continuing the same positive sign indefinitely, the sign of the quadratic coefficient of the magnetostriction in $\delta$-Pu turns negative beyond $\approx 200$ K (see Fig. 1c). A negative sign indicates the onset of thermal excitations into a higher energy electronic configuration with a larger magnetic moment, but whose equilibrium atomic size ($V_2$) is now significantly smaller than that of the other configurations—as frequently encountered in intermediate valence compounds of Yb (see Fig. 2b)[22,23,25,26]. The highly non-monotonic temperature-dependence of the magnetostriction in $\delta$-Pu is indicative of at least three different $f$-electron configurations ($E_0$, $E_1$, and $E_2$) being relevant (shown schematically in Fig. 2c).

Our magnetostriction measurements of $\delta$-Pu are corroborated by thermal expansion measurements[41], which, while lacking information on magnetic moments of the $f$-electrons, convey more direct information concerning the change in atomic volume between different configurations. Low temperature thermal expansion measurements (Supplementary Fig. 2a, b) find the electronic contribution to the thermal expansion from itinerant carriers to be overwhelmed by phonons at temperatures above ~10 K, as has also been suggested on the basis of heat capacity measurements[17]. A low temperature thermal expansion dominated by phonons is further validated by the published temperature-dependent lattice constant data[41] (replotted in Fig. 1d). On considering the thermal expansivity curves (shown in Fig. 1e), which are obtained from a temperature derivative of a smooth curve fit to the lattice constant data, it becomes evident that not until $T \gtrsim 50$ K does a notable departure from the phonon contribution (magenta curve in Fig. 1e) become apparent. The non-phonon contribution to the thermal expansion therefore mirrors the form of the magnetostriction, revealing excitations between electronic configurations to be an equally impactful in both thermodynamic quantities.

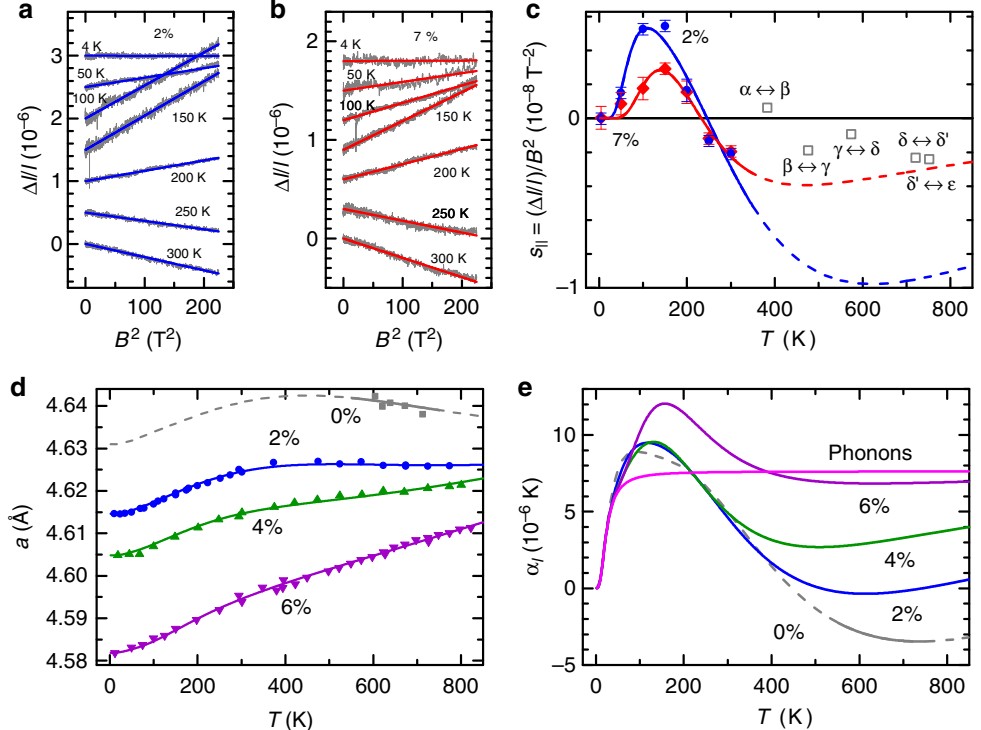

**Fig. 1** Magnetostriction and thermal expansion of Ga-stabilized $\delta$-Pu ($\delta$-Pu$_{1-x}$Ga$_x$). **a** Longitudinal magnetostriction $\Delta l/l$ (gray lines) of $x = 2\%$ versus $B^2$ at several different temperatures together with quadratic fits (blue lines), where $B$ is the magnetic field. **b** Longitudinal magnetostriction (gray lines) of $x = 7\%$ versus $B^2$ at several different temperatures together with quadratic fits (red lines). **c** Longitudinal magnetostriction coefficient $s_\parallel$ versus temperature for $x = 2\%$ (blue circles) and $x = 7\%$ (red diamonds). Error bars refer to the standard error of the mean, estimated by way of multiple sweeps. Open squares are the magnetovolume coefficient associated with each of the allotropic phase transitions in pure Pu, as described in the Results section. **d** Lattice parameter $a$ versus $T$ for $x = 0\%$ (gray squares), 2% (blue circles), 4% (green up triangles), and 6% (violet down triangles) from ref. [41]. **e** The thermal expansivity, obtained by applying $\alpha_l = a_{T=0}(\partial a/\partial T)$ to the fitted lines in **d**. The phonon contribution calculated for a Debye temperature of $\theta_D = 100$ K[17] is shown for comparison (magenta). The global fit to the data in **c** and **d** is discussed in the text and Methods, and is displayed using lines of the same color as the data points. Dashed lines indicate extrapolations

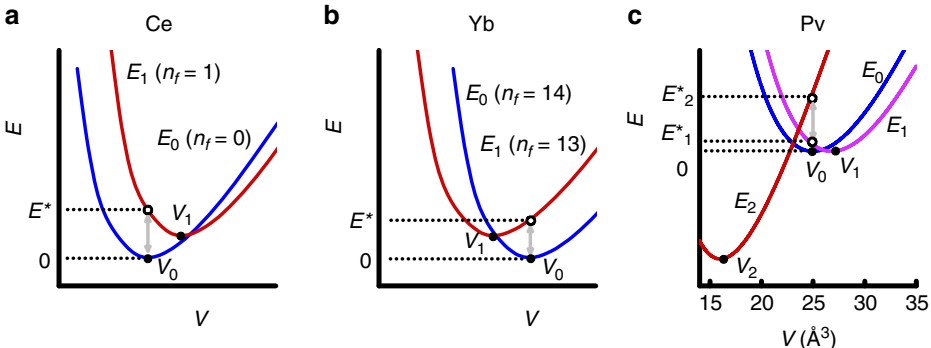

**Fig. 2** Schematic volume-dependent energies of different electronic configurations. **a** Schematic volume $V$-dependent energies $E$ for two configurations of Ce and its compounds, in which $E_0$ (blue line) has $n_f = 0$ 4$f$-electrons confined to the atomic core and $E_1$ (red line) has $n_f = 1$ 4$f$-electron confined to the core. Following Wohlleben[28] and Wittershagen and Wohlleben[60] we have adopted the convention whereby $E_1 > E_0$ at $V = V_0$ corresponds to $E^* > 0$ and $E_1 < E_0$ at $V = V_0$ corresponds to a $E^* < 0$. **b** A similar schematic for Yb. **c** A possible scheme for $\delta$-Pu that is consistent with $x$-dependent fits to magnetostriction and thermal expansion, consisting of three volume-dependent configurational energies $E_0$ (blue line), $E_1$ (violet line) and $E_2$ (red line) and two excitation energies $E_1^*$ and $E_2^*$ (indicated using dotted lines; see Methods). The number of 5$f$-electrons confined to the atomic core in each case remains to be determined

**Modeling of the free energy**. We proceed to establish validity of a multiconfigurational picture by first showing that the magnetostriction and thermal expansion in Ga-stabilized $\delta$-Pu are fully consistent with a model for the statistical thermodynamics of a multiple level system[22,28], and then, in the following section, by showing that the model accurately predicts the temperature and volume-dependence of heat capacity data. When two or more different electronic configurations with different energies coexist at a given value of the atomic volume $V$ (shown schematically in Fig. 2), their relative occupations can be described by a partition

function $Z_{\mathrm{el}}$, which produces an electronic contribution to the free energy of the form

$$F_{\mathrm{el}} = -k_{\mathrm{B}} T N \ln Z_{\mathrm{el}}. \tag{1}$$

Thermodynamic quantities, such as the quadratic-in-field magnetostriction coefficient $s_{||}$, thermal expansion coefficient $\alpha_l$ and heat capacity $C_V$, are then given by second derivatives $\left(\eta s_{||} \approx s_\nu = -\frac{\kappa_0}{B}\frac{\partial^2 F}{\partial \nu \partial B}, \quad 3\alpha_l \approx \alpha_\nu = -\kappa_0 \frac{\partial^2 F}{\partial \nu \partial T} \quad \text{and} \quad C_p \approx C_V = -T\frac{\partial^2 F}{\partial T^2}\Big|_V \right)$ of the total free energy $F(T, B, x) = F_{\mathrm{el}} + F_{\mathrm{ph}}$, where $F_{\mathrm{ph}}$ is the contribution from phonons (see Methods)[41,42].

We perform a simultaneous fit of $F(T, B, x)$ to $s_{||}(T, x) \approx \frac{1}{\eta B^2}[\nu_{\mathrm{el}}(T, B, x) - \nu_{\mathrm{el}}(T, 0, x)]$ and $a(T, x) - a_0(x) \approx \frac{a_0(x)}{3}\nu(T, 0, x) = \frac{a_0(x)}{3}[\nu_{\mathrm{ph}}(T, 0, x) + \nu_{\mathrm{el}}(T, 0, x)]$ in Fig. 1c, d using $\nu(T, B, x) \approx -K_0^{-1}\frac{\partial F(T,B,x)}{\partial \nu}\Big|_p$ (see Methods), where $K_0$ is the bulk modulus of the ground state configuration, $B = 15$ T in Fig. 1a, b, $a_0(x)$ is the lowest temperature value of $a$ for each Ga concentration $x$ and we assume $\eta = 3$ (see Methods)[23]. The electronic contribution

$$\nu_{\mathrm{el}}(T, B) \approx \frac{1}{Z_{\mathrm{el}}^*}\sum_{i=0,1,2}\sum_{\sigma=\pm\frac{1}{2}}\nu_i^* e^{-\frac{k_{\mathrm{B}}E_i^* + 2\sigma\mu_i^* B}{k_{\mathrm{B}}T}} \tag{2}$$

to the volume dilation $\nu(T, B)$ is derived in Methods, where

$$Z_{\mathrm{el}}^* = \sum_{i=0,1,2}\sum_{\sigma=\pm\frac{1}{2}} e^{-\frac{k_{\mathrm{B}}E_i^* + 2\sigma\mu_i^* B}{k_{\mathrm{B}}T}} \tag{3}$$

is the effective partition function (see also Supplementary Note 1). The phonon contribution $\nu_{\mathrm{ph}}(T)$ is obtained by numerical differentiation of $F_{\mathrm{ph}}$ (given in Methods). During fitting, the excitation energies $E_i^*$ (illustrated in Fig. 2c), the effective magnetic moments $\mu_i^*$ (assuming $\sigma = \pm\frac{1}{2}$ pseudospins) and effective equilibrium volume dilations or contractions $\nu_i^*$ for the different configurations $i$ are allowed to have independent values for $x = 2\%$ and $x = 7\%$ Ga concentrations; for other values of $x$, we assume $E_i^*$, $\mu_i^*$ and $\nu_i^*$ to interpolate linearly as a function of $x$

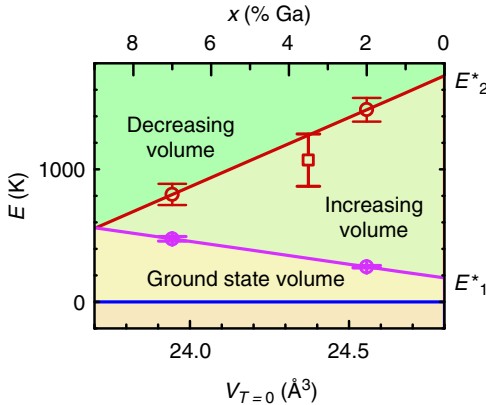

**Fig. 3** Schematic energies and volumes of Pu. Excitation energies $E_1^*$ (magenta circles) and $E_2^*$ (red circles) for $\delta$-Pu$_{1-x}$Ga$_x$ for $x = 2$ and 7% (upper $x$ axis) according to Table 1 (circles), with the lines of the same color indicating interpolations (or extrapolations for $x = 0$) assumed during fitting (see Methods). The lower horizontal axis shows the approximate atomic volume $V_{T=0}$ of the ground state ($T = 0$). Shaded regions illustrate the relative volume changes accompanying the excitations. The open square corresponds to the resonance energy observed in neutron scattering measurements[39], where the error bar indicates its approximate width

**Table 1 Fitting parameters**

| Quantity | $x = 2\%$ Ga | $x = 7\%$ Ga | All $x$ | Units |
|---|---|---|---|---|
| $(1+\nu_1^*)V_0$ | 24.66 ± 0.02 | 24.37 ± 0.11 | | Å³ |
| $E_1^*$ | 265 ± 10 | 476 ± 18 | | K |
| $\mu_1^*$ | 1.7 ± 0.3 | 1.3 ± 0.3 | | $\mu_{\mathrm{B}}$ |
| $(1+\nu_2^*)V_0$ | 21.29 ± 0.01 | 23.71 ± 0.08 | | Å³ |
| $E_2^*$ | 1450 ± 90 | 810 ± 80 | | K |
| $\mu_2^*$ | 2.9 ± 1.0 | 3.8 ± 1.4 | | $\mu_{\mathrm{B}}$ |
| $\gamma$ | | | 0.50 ± 0.04 | - |

Values of the various parameters obtained on performing a least squares fit, including errors (estimated from their covariance). The corresponding parameters of the ground state of $E_0^* = \mu_0^* = 0$, while $V_0 = 24.57$ and 23.87 Å³ for $x = 2$ and 7%, respectively (estimated from diffraction measurements)[41]

(or extrapolate linearly in the case of $x = 0$ in Fig. 3). Meanwhile, following the experimental finding that the Debye temperatures characterizing the heat capacities of $x = 2\%$ and $x = 7\%$ Ga-stabilized $\delta$-Pu (see Supplementary Fig. 3) and $x = 5\%$ Al-stabilized $\delta$-Pu[17] have very similar values of $\Theta_{\mathrm{D}} \approx 100$ K, we assume $\nu_{\mathrm{ph}}(T)$ to be independent of $x$, enabling us to reduce the number of fitting parameters. The results of the fitting are shown in Fig. 3 and Table 1. Adding confidence to this procedure, $E_1^*$ and $E_2^*$ are found to remain robust on restricting the fit only to $s_{||}(T, x)$ or, alternatively, on expanding the fit to include the magnetic susceptibility; the latter requiring substantially more parameters (see Supplementary Note 2 and Supplementary Tables 1 and 2 and Supplementary Fig. 4).

**Heat capacity measurements.** We proceed to show that the above form for $F(T, B, x)$, having been fit to $s_{||}(T, x)$ and $a(T, x)$, successfully predicts a third thermodynamic quantity; namely the heat capacity. On computing the heat capacity in Fig. 4a, we find multiconfigurational excitations to add ~5 Jmol$^{-1}$K$^{-1}$ to the heat capacity at $T \gtrsim 100$ K, bringing it into close agreement with the published experimental curve[17]. Multiconfigurational electronic excitations therefore produce the largest contribution to the heat capacity and entropy after phonons (see Fig. 4a, b), with the characteristic energy $E_1^*$ dominating at temperatures between $\approx 100$ and 300 K and $E_2^*$ coming in at higher temperatures (see Fig. 2c). Importantly, the entropy associated with the electronic excitations (see Fig. 4b) is more than sufficient to account for the $\approx 0.8 \times R\ln 2$ excess entropy previously identified as favoring the stabilization of $\delta$-Pu over $\alpha$-Pu at high temperatures (where $R = k_{\mathrm{B}}N_{\mathrm{A}}$ and $N_{\mathrm{A}}$ is Avogadro's number)[18,19].

A particularly striking finding is that the excitation energies $E_1^*$ and $E_2^*$ change rapidly as a function of the Ga content $x$ (plotted in Fig. 3), and in opposite directions. One predicted consequence of their rapid change with $x$ is that the heat capacity is expected to become strongly dependent on the Ga composition (see Fig. 4), thus providing a means for the extreme dependences of $E_1^*$ and $E_2^*$ on $x$ to be robustly verified by experiment. To confirm that the extreme sensitivity of $E_1^*$ and $E_2^*$ to $x$ and $V$ is a genuine effect, we calculate the heat capacity as a function of $T$ at different values of $x$ (see Fig. 4c) and compare it against an independent set of $x$-dependent heat capacity measurements (raw data contained in Supplementary Fig. 3). On taking the difference between the calculated heat capacity for $x = 2$ and 7% (from Fig. 4c), we find that it indeed accurately predicts the difference in heat capacity observed experimentally (shown in Fig. 4d), including both the absolute magnitude of the difference and the existence of a sign change in the difference at $\approx 130$ K. Since the primary effect of Ga substitution is to reduce the volume $V$ of $\delta$-Pu (the ground state atomic volume of $\delta$-Pu$_{1-x}$Ga$_x$ being shown on the lower

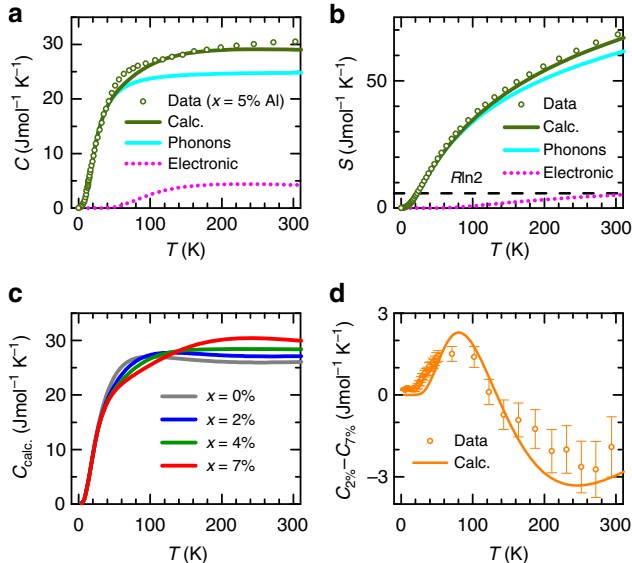

**Fig. 4** Heat capacity and Entropy of Ga-stabilized δ-Pu (δ-Pu$_{1-x}$X$_x$), where X = Ga or Al. **a** Calculated heat capacity versus temperature T for x = 5% (solid green line) compared against experimental heat capacity C data (for X = Al and x = 5% (green circles)[17]. The separate calculated phonon (cyan solid line) and electronic (magenta dotted line) contributions are also shown. We neglect possible differences that may exist between X = Al and Ga. **b** A comparison of the calculated and measured entropy of the same composition, where $S = \int_0^T (C/T^\dagger) dT^\dagger$. **c** The calculated heat capacity C$_{calc.}$ for different values of x in different colors, as indicated. **d** The calculated difference in heat capacity between x = 7% and x = 2% (orange line), compared against the experimentally measured difference (orange circles) together with error bars (corresponding to the standard error of the mean)

horizontal axis of Fig. 3)[36], we attribute the opposite variations of $E_1^*$ and $E_2^*$ with x to their sensitivity to volume, illustrated in Fig. 2c.

## Discussion
We have therefore uncovered two previously hidden electronic energy scales giving rise to significant entropy in excess of the Dulong-Petit value[20] of ≈25 Jmol$^{-1}$K$^{-1}$ at temperatures above the Debye temperature in plutonium. The strong variations of $E_1^*$ and $E_2^*$ with x shed light on the long unresolved questions[2] of why the volume collapse occurs and why it is inhibited by Ga substitution[36]. A likely energetic motivation for the volume collapse at low temperatures is provided by the steep decline in $E_2^*$ with decreasing volume (see Fig. 3a), which suggests an equilibrium volume ($V_2$) for $E_2$, similar to that found by density functional theory[12], that is lower in energy than that of $E_0$ in Fig. 2c. In practice, the volume collapse (which takes place via a series of steps in pure Pu)[2,37] is accompanied by additional translational symmetry breaking into the α phase (or α' phase in the alloys), which give rise to an energy barrier between the different volume phases[19]. In pure δ-Pu, $E_2^*$ is too high (≈1700 K according to an extrapolation to x = 0% in Fig. 3) to supply sufficient entropy to δ-Pu for it to remain stable over an extended range of temperatures, resulting in its ultimate collapse into the α phase. Conversely, in heavily substituted δ-Pu$_{1-x}$Ga$_x$, $E_1^*$ and $E_2^*$ are both sufficiently low to afford the δ phase in the alloy a significantly elevated entropy relative to that in pure plutonium, as evidenced by the higher electronic heat capacity above ≈130 K of heavily Ga-stabilized δ plutonium relative to its Ga-reduced counterpart. The additional entropy associated with the rapid descent of $E_x^*$

with increasing x thus provides δ-Pu$_{1-x}$Ga$_x$ with some degree of protection against collapsing into the α phase (see Methods)[19,37].

The strong interplay between the excitation energies and V is further demonstrated by the similarity in behavior of the magnetostriction to the transitional magnetovolume coefficient associated with each of the crystallographic phase transitions in plutonium[35]. We estimate the transitional magnetovolume coefficient, $s_{||}^{mv} \approx \frac{\mu_0 \kappa_0}{6} \Delta\chi/\Delta\nu$ (open squares in Fig. 1c), of pure Pu from the ratio of the previously measured jump $\Delta\chi$ in the susceptibility to the jump $\Delta\nu$ in volume dilation (plotted in Supplementary Fig. 5) at each of the phase transitions[2]. We find the magnetovolume and magnetostriction coefficients to be of similar magnitude and to exhibit similar positive-to-negative trends with increasing temperature, implying that $E_1^*$ and $E_2^*$ respond similarly to reductions in V caused by phase transitions as they do to reductions in V caused by Ga substitution.

Our findings shed light on the electronic structure of plutonium and its relation to other actinides and to rare earths. While energy scales of comparable magnitude to the larger excitation energy ($E_2^*$ in Fig. 3a) have been inferred from neutron scattering experiments (for x = 3.5%[39] plotted in Fig. 3a) and from fits made solely to the thermal expansion[41], their origins have remained controversial[39,43] and their volume-dependences have remained unknown. Electronic structure calculations have shown that Pu is able to exist in a larger number of near degenerate configurations than most other f-electron systems[29,31,33,44,45], (see e.g. Supplementary Fig. 6) with each having a different number of 5f-electrons confined or localized within the atomic core and different values of the equilibrium atomic volume, thereby providing a likely origin for $E_0$, $E_1$, and $E_2$ (shown schematically in Fig. 2c). The negative magnetostriction accompanying the negative thermal expansion at elevated temperatures implies that $E_2$ corresponds to an electronic configuration with fewer 5f-electrons confined to the atomic core but with a substantially larger effective magnetic moment; raising the interesting possibility of a very strong magnetic field being utilized to induce a volume collapse in δ-Pu.

Finally, we estimate the magnitude of the virtual valence fluctuations in δ-Pu, which we have found to have little to no role in its thermodynamic stabilization. On incorporating the valence fluctuation temperature $T_{fl}$ phenomenologically into $F(T, B, x)$[22,28], we find that $T_{fl} \lesssim 50$ K $\ll E_1^* < E_2^*$ (see Methods, and also Supplementary Note 2 and Supplementary Table 1), thereby justifying our ability to neglect $T_{fl}$ during fitting. However, our experimentally determined upper limit for $T_{fl}$ is at least an order of magnitude smaller than the effective Kondo temperature of ~$10^3$ K for δ-Pu suggested by contemporary electronic structure methods[5,8,34]. In the present model, $T_{fl} \sim 10^3$ K would mostly prevent the temperature and magnetic field from being able to change the occupancies of the different electronic configurations for the entire solid state regime $T \lesssim 10^3$ K, rendering $s_{||}$ and the electronic contribution to $\alpha_l$ small and largely temperature-independent over the entire range. A smaller energy scale for $T_{fl}$ can be more easily reconciled with the conventional linear-in-T Sommerfeld contribution that persists to only ~20 K in heat capacity experiments[17], which justifies our being able to neglect its contribution to the entropy at higher temperatures.

Our finding of a small $T_{fl}$ energy scale implies that the virtual valence temperature and excitation energies of Pu's largest volume phase, δ-Pu, are similar in magnitude to those ($T_{fl} \approx 40$ K and $E^* \approx -700$ K)[28] of the largest volume phase of cerium, γ-Ce[27]. Possible explanations for the similar magnitudes are that the degree of overlap of the f-electron wave functions is similarly weak in the largest volume phases of both Ce and Pu[27,28], or that both are similarly close to being of integer valence. In Ce, $T_{fl}$ only becomes significantly larger upon undergoing a considerable

collapse in volume into the $\alpha$-Ce phase, suggesting that this is likely also to be the case when Pu undergoes a volume collapse into the $\alpha$ phase. Despite the entropy associated with a small $T_{fl}$ being similarly important for the stabilization of both $\delta$-Pu and $\gamma$-Ce at elevated temperatures (and also the high temperature phase of YbInCu$_4$)[46–49], these systems have differences in the arrangement of their electronic configurations that cause their behaviors to differ. In $\gamma$-Ce, the high-temperature phase is stabilized by the large entropy contribution of a lowest energy magnetic 4$f$-shell configuration (hence the negative value of $E^*$)[28], with the excitations between this magnetic configuration and a higher energy non-magnetic configuration being of lesser importance. In $\delta$-Pu, by contrast, the lowest energy 5$f$-electron shell configuration appears to be non-magnetic[35] (see Table 1), with the large contribution to the entropy instead originating from excitations to multiple higher energy configurations with magnetic moments.

## Methods

**Magnetostriction measurements.** Variations $\Delta l_{||}$ in the sample length $l_{||}$ along the direction of the magnetic field are measured either upon sweeping the temperature at zero magnetic field or on sweeping the magnetic field up to 15T and back to zero at fixed temperature, for both polarities of the magnetic field. The measurements are made using the fiber Bragg grating method[38], in which we record the spectral information on the light reflected by 1- and 2-mm-long Bragg gratings inscribed in the core of a 125-$\mu$m telecom-type optical fiber by means of a swept wavelength Hyperion Interrogator manufactured by Micron Optics. A flat face of a sample is attached to a single grating on its own fiber using cyanoacrylate glue. One or two empty gratings on the same fiber provide a means for compensating for the temperature-dependence of the diffraction index of the fiber in the absence of a sample.

Multiple fibers are fed through stainless steel capillary tubes into a brass can that forms the body of the sample primary encapsulation. Using this method, multiple samples can be co-encapsulated, while a steel hepa filter enables $^4$He gas or liquid to circulate. The fibers holding the samples are anchored to a metallic block for thermalization, made of non-magnetic stainless steel in the case of samples 1 and 2 and copper in the case of samples 3 and 4. Thermometers are also anchored to the metallic block inside the can. The brass can is then mounted on the end of a probe inside a secondary containment containing either vacuum or $^4$He, which can be pumped through a high through-put hepa filter situated on the pumping line. The secondary containment, which has its own thermometer, is then placed inside a variable temperature insert (VTI) that itself goes inside the bore of a 15T superconducting magnet.

The temperature is controlled via the VTI by using a heater and also, when necessary, using a secondary heater on the secondary containment. Using this arrangement, the temperature can be stabilized to $\approx$50 mK, with a small thermal drift occurring over timescales of order several hours. To eliminate the effect of thermal drift during magnetostriction measurements, up and down sweeps of the magnetic field are averaged and the temperature adjusted accordingly. Negative and positive sweeps of the magnetic fields are also compared to ensure reproducibility of the result.

Magnetostriction measurements are obtained on four different samples (see Supplementary Fig. 1). A higher signal-to-noise ratio is observed in the case of samples 1 and 2, which we therefore use for performing fits. Samples 3 and 4 are found to have magnetostriction coefficients that are consistent with the fits to samples 1 and 2. Error bars (s.e.m.) are estimated after repeating the magnetostriction measurements at the same temperature, often with a different polarity of the magnetic field.

**Sample preparation details.** All of the samples have an isotopic composition of 0.02% $^{238}$Pu, 93.6% $^{239}$Pu, 5.9% $^{240}$Pu, 0.44% $^{241}$Pu, and 0.04% $^{242}$Pu with regards to Pu[50]. Prior to our measurements on the $\delta$-Pu$_{1−x}$Ga$_x$, the $x$ = 2% material had aged 11 years since casting whereas the $x$ = 7% material had aged 7 years since casting. The gallium concentration does not vary significantly from sample to sample, but does tend to congregate in the center of grains. We employ a thermal treatment, which includes a vacuum homogenization process, that makes the Ga uniform through the sample and also partially 'resets' the effects of aging. The thermal treatment eliminates stresses and some accumulated impurities (notably H), but does not remove the radiolytically generated impurities like U, Am, and He. In the case of $\delta$-Pu$_{0.98}$Ga$_{0.02}$, the samples are processed at 450 °C under vacuum for $\approx$100 hours to achieve homogenization, while in the case of $\delta$-Pu$_{0.93}$Ga$_{0.07}$, the sample is processed at 525 °C under vacuum for $\approx$ 50 hours to achieve homogenization. This process produces polycrystalline material that is low in H and has reduced Fe. Typical impurities, such as Al and Fe are at the 100 parts per million level.

Several polycrystalline samples with $x$ = 2% and 7% are prepared in the form of plates of a few millimeters with masses ranging between 16 and 40 mg. Samples 1

and 2, measured in the main text, have Ga compositions of 2 and 7% with masses of 16.2 mg and 30.7 mg, respectively and dimensions on the order of $\sim$1 mm $\times$ 4 mm with a thickness of 150 $\mu$m. The samples are lightly polished prior to gluing onto the fibers in order to remove any possible surface oxidation. The glue also has the effect of protecting the measured flat surfaces of the samples against oxidation during their loading into the VTI.

For the $x$ = 2% sample, the sample length is observed to drift slowly in time when the temperature is set close to $\approx$150 K as a consequence of the partial and gradual transformation of the $\delta$ phase into the $\alpha'$ phase of plutonium, where the $\alpha'$ phase in Ga substituted Pu has the same structure as the $\alpha$ phase in pure Pu. The total change experienced during the course of the stabilization at $\approx$ 150 K is $\Delta l/l \approx −0.15\%$, which, given the smaller atomic volume of $\alpha$-Pu, corresponds to 2.3% of the sample (by volume) in contact with the fiber transforming. No similar transformation is observed on measuring the $x$ = 7% sample.

**Thermodynamics.** The coefficients of thermal expansion and volume magnetostriction are given by[24]

$$\frac{\partial \nu}{\partial T} = \alpha_\nu = -K_0^{-1} \frac{\partial^2 F}{\partial \nu \partial T} \quad \text{and} \quad s_\nu B = \lambda_\nu = -K_0^{-1} \frac{\partial^2 F}{\partial \nu \partial B}, \quad (4)$$

respectively, where $F$ is the free energy, $T$ is temperature, $B$ is the magnetic field, $K_0$ is the bulk modulus and $\nu = (\Delta V/V_0)$ is the volume expansion (or contraction). In the absence of broken time reversal symmetry (e.g., as for a ferromagnetic and some types of non-collinear antiferromagnetic ground state)[51], $\lambda_\nu(B)$ is linear in magnetic field, in which case the volume increases quadratically with field with the coefficient $s_\nu = \lambda_\nu/B$.

We assume the free energy to be the sum $F = F_{el} + F_{ph}$ of electronic and phonon contributions. The phonon contribution is given by[41,42]

$$F_{ph} = Nk_B T \left[ \frac{8}{9} \frac{\Theta_D (1 + \nu)^{-\gamma}}{T} + 3 \ln \left[ 1 - e^{-\frac{\Theta_D (1+\nu)^{-\gamma}}{T}} \right] - D \left( \frac{\Theta_D (1 + \nu)^{-\gamma}}{T} \right) \right] \quad (5)$$

where $N$ is the atomic density (inverse of the unit cell volume), D($x$) is the Debye function and $\gamma \approx 0.5$[52].

For systems with multiple electronic configurations that have the potential to coexist[53], we assume that each configuration $i$ has its own unique energy $E_i(\nu)$ that depends on $\nu$ in the manner illustrated in Fig. 2, and as predicted to be the case in plutonium[29,33]. The multiconfigurational partition function in Eq. (1) can then be written in the form

$$Z_{el} = \sum_{i=0,1,2} \sum_{\sigma = \pm \frac{1}{2}} e^{-\frac{k_B E_i(\nu) + 2\sigma \mu_i^* B}{k_B T}}. \quad (6)$$

Note that the summation is made over configurations that have different functional forms for $E_i(\nu)$, but are always at the same volume $V$. $V_i$ refers to the 'equilibrium volume' at which a given configuration would be located, were it to have the lowest energy at $T = 0$.

The multiple configurations consist of states in which different numbers $n_f$ of $f$-electrons are confined to the atomic core, or different crystal electric field levels in which the same number of $f$-electrons are confined to the atomic core[22,28]. However, the latter are typically more closely spaced in energy and volume. To minimize the number of fitting parameters, we assume an effective moment $\mu_i^*$, which refers either to that of the lowest crystal electric field level or an average over two or more occupied levels for a given value of $n_f$. The Van Vleck contribution can also add to $\mu_i^*$, as this is known to vary as a function of $n_f$.

**Numerical simulations and fitting procedure.** To facilitate fitting to experimental data[22,28], we first differentiate $F$ with respect to the $\nu$ to obtain

$$\nu(T, B) \approx \int_0^T \alpha_\nu(T'', B) dT'' = -K_0^{-1} \frac{\partial F}{\partial \nu}|_p, \quad (7)$$

where, here, $K_0$ refers the bulk modulus of the ground state configuration. For the phonon contribution, we proceed to calculate its contribution ($\nu$) numerically, while for the electronic contribution, differentiation yields the conveniently trivial result given in Eqs (2) and (3). Since the overall extent of the volume expansion in Fig. 1d is $\nu \lesssim 0.6\%$ for $x$ = 2% and $\nu \lesssim 2\%$ for $x$ = 6% samples, we have simplified the fitting procedure by setting $\nu$ to zero on the right-hand-side of Eq. (2) after differentiating. Following through with this approximation amounts to neglect of a possible 4 K temperature-dependent shift in $E_i$ for $x$ = 2% and a possible 30 K shift for $x$ = 7%. The changes in $E_i$ with $T$ are significantly less than the experimental uncertainty for $x$ = 2% and comparable to the experimental uncertainty for $x$ = 7% (typical error bars listed in Table 1). By comparison, volume changes induced by a magnetic field are only of order 1 ppm. Setting $\nu = 0$ on the right-hand-side simplifies the fitting procedure by allowing us to adopt effective parameters: $\nu_i^* = Nk_B K_0^{-1} \times \frac{\partial E_i}{\partial \nu}$ and $E_i^* = E_i(\nu = 0)$ in Eqs (2) and (3).

Provided $V$ is sufficiently close to the minimum of a $E_i(\nu)$ curve at $V = V_i$ in Fig. 2, one can then use a parabolic approximation:

$$E_i(\nu) = E_{i,0} + \frac{1}{2} K_i (\nu - \nu_i)^2 / k_B N, \quad (8)$$

where $K_i$ is the bulk modulus of the configuration and $\nu_i = (V_i - V_0)/V_0$ is the

relative volume dilation at which it has its lowest energy. In this case, $\nu_i^* = \nu_i \left( \frac{K_i}{K_0} \right)$. It is important to emphasize, however, that the volume dilation parameter $\nu_i^*$ obtained from fitting is not the actual volume dilation associated with the equilibrium volume of a given valence state, but, rather, a renormalized volume dilation parameter, which limits our ability to make accurate estimates of the equilibrium volume of each of the excited valence states in $\delta$-Pu. However, this has no discernible impact on the calculations of the heat capacity and entropy.

Since $\delta$-Pu is both cubic and polycrystalline, $1 \lesssim \eta \lesssim 3$ [23], although its precise value has no bearing on the entropy or heat capacity. The leading order quadratic form of the magnetostriction generally arises from the cancellation of the odd terms in the partition function upon summing the spin up and down pseudospin components. On substituting different values of $B_{max}$ in the magnetostriction coefficient numerical simulations, we find no significant deviation from a conventional quadratic form in the model.

A least squares fit is performed simultaneously to both the quadratic magnetostriction coefficient and the thermal expansion volume of $\delta$-Pu$_{1-x}$Ga$_x$, with the minimization being made with respect to the product of the sum of the squares of both quantities. During fitting, we assume a fixed Debye temperature of $\Theta_D = 100$ K [17], but leave $\gamma$ as a free parameter. All fitted parameters are listed in Table 1. As a demonstration of self consistency of the fitted model, the activation energy $E_2^*$ of the upper excited electronic configuration and the magnitude of the change in its characteristic volume are both found to increase on reducing the amount of Ga (see Table 1). Conversely, the activation energy $E_1^*$ of the lower excited valence configuration and the magnitude of the change in its characteristic volume are both found to decrease on reducing the amount of Ga.

**Virtual valence fluctuations.** It has been shown that virtual valence fluctuations can be phenomenologically modeled [22,28] by introducing an effective $T' = (T^m + T_{fl}^m)^{\frac{1}{m}}$ that is substituted in place of $T$ in the electronic contribution to the free energy, and thermodynamic derivatives thereof. Here, $T_{fl}$ is the virtual valence fluctuation temperature, or Kondo temperature, while $m = 1$ or 2. In typical rare earth intermediate valence systems, there are only two relevant configurations that need to be taken into consideration (e.g. Fig. 2a, b). On substituting $T'$ in place of $T$ in our fits to $s_{||}$ and $a$ in Fig. 1, we find that $T_{fl} \to 0$, rendering it not useful as a fitting parameter. However, provided $m = 2$, reasonable looking fits to the data can still be obtained using fixed values of $T_{fl} \lesssim 50$ K, thus providing an upper bound for $T_{fl}$ in $\delta$-Pu.

**Heat capacity measurements.** Heat capacity measurements (raw data in Supplementary Fig. 3) are made on samples of $\delta$-Pu$_{1-x}$Ga$_x$ of mass $\approx 5.5$ mg and $\approx 4.1$ mg for $x = 2$ and 7%, respectively, with small samples being chosen to minimize the effect of self heating. The measurements are made in a standard Quantum Design physical properties measurement system (PPMS), with the same addendum (and a similar amount of grease) being used each time. While the addendum, inclusive of Apiezon N grease, is measured each time prior to sample mounting, a small difference in the amount of grease for each sample introduces an additional experimental uncertainty. It should be noted, however, that the heat capacity of Apiezon N grease consists of a sharp peak at $T \approx 300$ K [54], which is quite different from the form of the difference $C_{2\%} - C_{7\%}$ in Fig. 4d.

The differences in Pu and Ga content in each of the samples also introduces a systematic error in the difference, and the extent to which this difference can be attributed to the electronic contribution. Since the contribution to the heat capacity from phonons universally saturates at the Dulong-Petit value of $\approx 25$ Jmol$^{-1}$K$^{-1}$ regardless of the Pu content, the subtracted quantity in Fig. 4d is free from any significant phonon contribution above $\approx 50$ K. The accuracy of the remaining $\approx 5$ Jmol$^{-1}$K$^{-1}$ electronic contribution in each sample is affected by $\Delta x = 5\%$ differences in Pu and Ga content, therefore making the systematic error in making the subtraction $5\% \times 5$ Jmol$^{-1}$K$^{-1}$ = 0.25 Jmol$^{-1}$K$^{-1}$. However, this is significantly less than the measurement error bar in Fig. 4d.

**Relevance of fitting results to the electronic structure.** The observed changes in magnetostriction with increasing temperature indicate that the magnetic moment appears to be smallest for the ground state configuration (see Methods), suggesting its possible correspondence to $n_f = 4$ or 5 5$f$-electrons confined to the atomic core [29,33,55]. Both of these configurations have the potential for orbital compensation to produce small moments [31,56] compatible with the absence of magnetic ordering [35]. Partial occupancy of both $n_f = 4$ and 5 has further been suggested on the basis of neutron scattering structure factor measurements [39], although such measurements are performed at a temperature of $T = 293$ K that is sufficiently high for both to be thermally occupied.

**Energy level schematics.** Cohesion in metals is generally expected to give rise to an energy $E$ versus linear dimension of the form [57]

$$E = \left( a_0 - \frac{a_1}{a} + \frac{a_2}{a^2} \right), \qquad (9)$$

where, here, $a = (4V)^{\frac{1}{3}}$ refers to the lattice parameter and $a_0$, $a_1$ and $a_2$ are constants. For the schematics in Fig. 2a–c, the $E$ versus $V$ curves are assumed to have

this form. We perform a fit of Eq. (9) to the calculated energy for $\delta$-Pu of Svane et al. [33], confirming that Eq. (9) is approximately valid for electronic structure calculations of plutonium (see Supplementary Fig. 6). The minima occur at $a_{min} = \frac{2a_2}{a_1}$, with the bulk modulus at $a = a_{min}$ being given by $K = \frac{\partial^2 E}{\partial \nu^2} \big|_{a = a_{min}} = \frac{1}{72N} (a_1^8 / a_2^7)$. In Fig. 2c, the $E$ versus $V$ schematic has been calculated using Eq. (9) so that $E_2 - E_0$ and $E_1 - E_0$ are consistent with $E_1^*$ and $E_2^*$ in Fig. 2d, respectively. Only the bulk modulus $K_0$ (at $a = a_{min}$) associated with $E_0$ has been measured directly. For $E_1$ and $E_2$, we have arbitrarily assumed $K_1 = 40$ GPa and $K_2 = 50$ GPa for $E_1$ and $E_2$, respectively.

The rapid fall of $E_2^*$ with decreasing volume suggests its minimum is located at a volume and energy that is significantly lower than that of $E_0$, which is consistent with $E_0$ being representative of a metastable configuration separated from a lower energy configuration by an energy barrier [36,37]. The volumes of $\alpha$ and $\alpha'$ are comparable to the equilibrium volume of $E_2$ in Fig. 2c, suggesting that the structural transformation to $\alpha$ may be a secondary effect associated with the volume collapse. For the volume collapse to occur, the net energy gain associated with the transition needs to be equal to or greater than the energy losses associated with what are essentially displacive structural transitions from $\delta$ to $\alpha$ ($\alpha'$). In Pu$_{1-x}$Ga$_x$, the transitions from $\delta$ to $\alpha$ ($\alpha'$) always occurs for $x \lesssim 2\%$.

**Effect of Ga on the relative $\alpha$ and $\delta$ phase stability.** The energies $E_1^*$ and $E_2^*$ in Fig. 2a primarily determine how the entropic contribution $-TS$ to the free energy of $\delta$-Pu$_{1-x}$Ga$_x$ changes in response to temperature and composition $x$. A dominant role played by the entropy is suggested by the similarity of the rate $\partial E_2^* / \partial x \approx -125$ K per% at which $E_2^*$ descends with increasing Ga concentration $x$ to the rate $\partial T / \partial x \sim -100$ K per% at which the transition exhibiting the largest volume reduction and largest enthalpy descends in temperature with $x$. In pure Pu, the largest volume reduction and 75% of the total enthalpy on transforming between $\delta$ and $\alpha$ occurs at the $\beta$ to $\alpha$ transition, located at $\approx 398$ K [58] (the $\delta$ phase itself is stabilized at $\approx 593$ K). In Pu$_{1-x}$Ga$_x$ and for $x = 1\%$, a direct $\delta$ to $\alpha'$ transformation takes place on cooling at $\sim 230$ K, while for $x = 2\%$, a similar transformation occurs at $\sim 150$ K [36]. The smallness of the volume fraction transforming in the case of $x = 2\%$ (see sample preparation details) suggests that the transformation occurs very sluggishly at ambient pressure, with the temperature of $T = 150$ K being barely sufficient to overcome the energy barrier separating $\delta$ and $\alpha'$ phases.

In order to determine whether a phase transformation occurs between $\delta$-Pu$_{1-x}$Ga$_x$ and $\alpha'$-Pu$_{1-x}$Ga$_x$ (or $\alpha$-Pu in the case of $x = 0$), it is also necessary to know the difference in internal energy $E$ between these two phases in the limit $T \to 0$. In the case of $\delta$-Pu$_{1-x}$Ga$_x$, its internal energy is given by the minimum of $E_0$ in Fig. 2c. In the case of $\alpha$, one possibility is that its internal energy as a function of volume resembles $E_2(V)$ in Fig. 2c; this should be considered with caution, however, since $E_2$ is constructed on the basis of several assumptions: the value of the bulk modulus, the validity of Eq. (9) and equivalency between $x$ and volume.

It has been shown experimentally that whereas the volume of $\delta$-Pu$_{1-x}$Ga$_x$ decreases with increasing $x$, the volume of $\alpha'$-Pu$_{1-x}$Ga$_x$ increases with increasing $x$ [36]. $E_0(V)$ must therefore move to smaller volumes with increasing $x$, while the internal energy versus volume curve for $\alpha'$-Pu$_{1-x}$Ga$_x$ [perhaps resembling $E_2(V)$] must move to larger volumes. The difference in internal energy between the $\delta$-Pu$_{1-x}$Ga$_x$ and $\alpha'$-Pu$_{1-x}$Ga$_x$ phases is therefore likely to decrease with increasing $x$, thereby reducing the energetic favorability of the $\alpha'$ over the $\delta$ phase.

The reduction in internal energy between the $\delta$ and $\alpha'$ phases with increasing $x$ can be understood by the following argument: the movement in the minima of $E_0(V)$, $E_1(V)$, and $E_2(V)$ with increasing $x$ must ultimately take place via gradual changes in the parameters $a_0$, $a_1$, and $a_2$ in Eq. (9) with increasing $x$. Since the differences between $E_0(V)$, $E_1(V)$ and $E_2(V)$ for the $\delta$ and $\alpha'$ (or $\alpha$) phase are determined primarily by differences in the electronic configurations of the 5$f$-electron shell, these differences become diluted as Pu within the lattice is substituted with Ga. They must ultimately vanish at $x = 1$.

Valence fluctuations are another factor that will affect the phase stability of the $\alpha$ ($\alpha'$) phase. While valence fluctuations are insignificant in the $\delta$ phase, they are likely to become very significant in the $\alpha$ phase due to an increase in the overlap of $f$-electron wave functions. In Ce, for example, the valence fluctuation temperature is reported to reach between 2000 and 5000 K (depending on the applied pressure) within its smallest volume $\alpha$ phase [27,28]. Increased dynamical instability of the $\delta$-phase lattice with increasing $x$ is another possible factor affecting phase stability [59], although the energies involved appear to be small compared to $E_1^*$ and $E_2^*$.

## Data availability
The data that support the findings of this study are available from the corresponding author upon reasonable request.

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

## Acknowledgements

The work was performed under the Los Alamos National Laboratory LDRD program: project "20180025DR." Measurements were performed at the National High Magnetic Field Laboratory, which is supported by the National Science Foundation, Florida State and the Department of Energy. N. H. thanks John Wills, Jianxin Zhu, Angus Lawson, Jason Lashley, Albert Migliori, Boris Maiorov, and John Joyce for insightful discussions.

## Author contributions

N.H., J.B.B., M.R.W. and M.J. performed the measurements. P.H.T. and S.R. prepared and mounted the samples. M.J., J.B.B., M.R.W. and F.F.B. developed the experimental apparatus. N.H. performed the modeling. N.H., P.H.T. and M.J. wrote the manuscript. P. H.T. arranged all of the sample transportation logistics.

## Additional information

**Competing interests:** The authors declare no competing interests.

