## [Peer Review File · Nature Communications]

Reviewers' comments:

Reviewer #1 (Remarks to the Author):

This paper reports significant contribution of a degree of freedom of electric configurations of 5f electrons to the large entropy that stabilizes the delta-phase of plutonium. The key finding of this work is the three different electron-configuration states whose energies are E_0 , E_1 , and E_2 , respectively. These states enable the system to behave nonmonotonic temperature dependence of the magneto-volume striction. Also, the volume collapse in pure delta-Pu to alpha phase is explained in terms of a large energy relaxation of E_2 state with the much smaller atomic volume V_2 . It is interpreted that the stabilization of the phase owing to the entropy is more significant by decreasing (increasing) of E_2^* (E_1^*) with decreasing the volume. A decrease in the volume is caused by several reasons, i.e., decreasing temperature in low temperature region, phase transition to another phase, Ga(Al) substitution, and applying magnetic field at relatively high temperatures. The similarity in behavior of the magnetostriction to the transitional magnetovolume coefficient associated with each of the crystallographic phase transition can support the suggested volume dependence of the excitation energies (E_1^* and E_2^*). I feel that the discussions are mostly fair and the values of fitting parameters obtained look reasonable. Since the anomalous temperature dependence of the volume magnetostriction has never seen in other systems, it can be one of characteristic features of actinoids and deepen our understanding of 5f-electron systems. I think that the paper is worth publishing.

However, I have several comments as follows.

(1) I feel that the structure (or an order of description) of the paper can be improved. I don't agree it is a good idea showing the x -dependence of the excitation energies in the very first figure, i.e., in Figure 1a. It is more natural to show this figure after proposing a model of electronic configurations and analyzing the experimental results.

(2) One of the conclusions of this paper is that the Kondo effect or valence fluctuation effect is not very important for understanding the unusual temperature dependence of the volume magnetostriction. However, authors also explain that a large electronic contribution to the lattice thermodynamics similar to that observed in this work often occurs in valence fluctuating system such as Ce or Yb. If the energy levels of more than two electronic configurations with different f -electron occupancy (n_f) are close each other, the quantum mechanical mixing tends to occur and valence fluctuation or mixing is generally observed. If the entropy becomes large because of the fact that the energy levels with different n_f are located close each other, it is likely that valence fluctuation also becomes significant. Actually, the behavior of the volume magnetostriction at low temperatures in Fig. 2c is suggested to be in a similar physical situation of Ce changing its valence from nonmagnetic valence fluctuating (alpha) phase to magnetic integer valence (gamma) phase with elevating temperature. The numerical fitting procedure indicated a low Kondo temperature compare to the excitation energies (E_1^* , E_2^*). However, I think, in order to understand the physical situation better, more explanation is needed why the valence fluctuation effect is not significant although the different n_f states locate close each other.

(3) Since the x -dependences of E_1^* and E_2^* are linear, it seems to be strange only 2% Ga doping suddenly stabilize the delta phase. Considering $F=E-TS$, where F is the free energy and E , T , S are internal energy, temperature, and entropy, respectively, the phase can be stabilized by increasing temperature at finite temperature. At $x=0$, delta-phase cannot be stable, because S is not large enough and an relaxation to the bottom of E_2 curve (Fig3c) is large. With finite x , the three energy levels close each other and an enhancement S saves the energy. However, it is likely that x -dependence of f E_1^* and E_2^* is small to realize the phase stability with relatively small x such as 2%-doping. I would like to ask authors give an explanation for this question.

(4) In page 6, from line 4 to line7: Not until $T \gtrsim 50$ K to the lattice constant data.

Does this sentence explain the behavior of curves shown in Fig 2e ? I cannot understand which graph is referred. Please revise it.

(5) I feel that Method part is too long compared to the main text. Outline of the fitting procedure can be included in the main text. It is because the fitting is essentially important to validate the conclusion of this paper. It would also be nice to include Table 1 in the main text. I think the fact that the magnetic moment of E2 (μ_{2*}) is larger than that of (μ_{1*}) explains the sign change of temperature variation of the volume magnetostriction. I also think that the obtained values except μ_{*} shown in Table 1 are worth explaining in the main text.

Reviewer #2 (Remarks to the Author):

In my view this manuscript cannot be accepted in its present form. I will explain my arguments below.

- The manuscript is written in an unclear style and is difficult to comprehend for someone only partially familiar with the techniques. For example figure 1a is put forward and cannot be understood without explanation, which is scattered in the manuscript. This should be corrected in my view.
- The introduction contains information that is common knowledge and does not need to be included here. Figures 1b and 1c present no new data or evidence. They can be found in other works.
- There is no information about the materials used for the measurements and their treatment. In this specific case this is important as Pu is a radioactive substance and the radioactive decay introduces changes in the materials. So information on the isotopic composition, age, thermal treatments are all required to assess the results of the manuscript.
- If I understand correctly, the phonon calculations have been made with a single value for the Debye temperature. This seems not correct to me, or at least I am missing the arguments why this can be done. The phonons are affected by the impurities (Ga, Al) and a change in the Debye temperature can be expected. Then also radiation plays a role. Lattice defects resulting from decay also affect the phonons, so the difference between the 2% and 7% Ga samples may be caused by other factors as well. In the approach employed here, everything is squeezed into an electronic contribution, whereas reality may be more complex.
- Figure 4 is not consistent with Figure 6. In the latter the 7% sample has a lower heat capacity than the 2% sample below 120 K, whereas in Figure 4d the difference is positive up to this temperature. I hope it is just a simple error in the presentation of both experimental and calculated data. Personally I doubt whether this difference is really significant, realising that the measurements were made on a sample of 1 mg. This is especially true at high temperatures, where the contribution from the puck becomes big compared to that of the sample. The authors claim that the difference is not affected by the subtraction of the addendum, but I am not convinced by this argument. Only if the same addendum curve can be applied, but then the same puck should be used. But the self-heating effect is slightly different, and I guess the amount of grease to fix the samples too. Therefore, the error of about 2 J K⁻¹ mol⁻¹ seems underestimated in my view.

Overall, interesting paper, interesting results, but in the end the conclusion is the result of fitting of a model, where for my feeling the neglect of other effects might lead to an overestimation of the electronic entropy.

Reviewer #3 (Remarks to the Author):

I read with high interest this paper on magnetostriction measurements in plutonium.

As a theoretician, I cannot judge some technical part of the paper. It seems however that measurements are precise and thus useful and represent an important contribution to the physics of plutonium.

I have three main comments :

1. The model used to explain the non monotonous magnetostriction effect is interesting. Indeed the description of temperature effect is very important.

Low temperature description of actinides can be obtained by various method, see Ref 13 and PRX 5 011008 (2015), and JPCM 30 405603 (2018) for recent works. These works present structural properties of phases of plutonium at 0 K.

These papers are in nice agreement with the thermodynamics at 0 K of phases of plutonium in the paper (PRB 58, 15433 (1998)).

So 0K properties are described by theories.

Moreover in this last paper by Wallace, it is underlined that electronic entropy effect and anharmonic effect cannot be disentangled.

This is also in agreement with the paper of Manley et al on phonons which reports an excess entropy. Would it be possible that this excess entropy comes from anharmonic phonon effect instead of electronic effect as proposed in the manuscript ? It is a key point that should be addressed in the current paper.

2. The comparison to Cerium is surprising. In cerium, there is a local moment in the gamma phase and it explains the important role

of the magnetic field and the entropy (see the cited paper of Drymiotis et al).

In delta plutonium, there is no magnetic moment (cf Ref 14). So it is a basically a different physics.

In cerium, it seems to me that the current explanation of entropy is just a fluctuation of the electron in the f orbitals

and not the valence fluctuation.

It should be commented in the paper.

3. A recent work JPCM 29, 245402 (2017) discuss the role of impurity on the dynamical stability of delta plutonium.

Are the conclusion of this paper in agreement with the vision proposed in the manuscript ?

In view of the question, it really seems that the paper is interesting, but I need clarification on these questions.

Moreover, the writing is somewhat unclear, in particular the conclusion could be rewritten to be more clear. In particular, do the virtual fluctuations are important or not ? Using shorter sentences, and clear ideas would be helpful for the reader.

Our response to the Reviewer comments are indicated below in **bold red font**. Changes to the main text (other than trivial typos) are highlighted in **violet**.

Reviewer #1 (Remarks to the Author):

This paper reports significant contribution of a degree of freedom of electric configurations of 5f electrons to the large entropy that stabilizes the delta-phase of plutonium. The key finding of this work is the three different electron-configuration states whose energies are E_0 , E_1 , and E_2 , respectively. These states enable the system to behave nonmonotonic temperature dependence of the magneto-volume striction. Also, the volume collapse in pure delta-Pu to alpha phase is explained in terms of a large energy relaxation of E_2 state with the much smaller atomic volume V_2 . It is interpreted that the stabilization of the phase owing to the entropy is more significant by decreasing (increasing) of E_2^* (E_1^*) with decreasing the volume. A decrease in the volume is caused by several reasons, i.e., decreasing temperature in low temperature region, phase transition to another phase, Ga(Al) substitution, and applying magnetic field at relatively high temperatures. The similarity in behavior of the magnetostriction to the transitional magnetovolume coefficient associated with each of the crystallographic phase transition can support the suggested volume dependence of the excitation energies (E_1^* and E_2^*). I feel that the discussions are mostly fair and the values of fitting parameters obtained look reasonable. Since the anomalous temperature dependence of the volume magnetostriction has never seen in other systems, it can be one of characteristic features of actinoids and deepen our understanding of 5f-electron systems. I think that the paper is worth publishing.

However, I have several comments as follows.

(1) I feel that the structure (or an order of description) of the paper can be improved. I don't agree it is a good idea showing the x-dependence of the excitation energies in the very first figure, i.e., in Figure 1a. It is more natural to show this figure after proposing a model of electronic configurations and analyzing the experimental results.

Response to (1): We have moved Fig. 1a so that it is now Fig. 3. Importantly, it's new location is after free energy model.

(2) One of the conclusions of this paper is that the Kondo effect or valence fluctuation effect is not very important for understanding the unusual temperature dependence of the volume magneto striction. However, authors also explain that a large electronic contribution to the lattice thermodynamics similar to that observed in this work often occurs in valence fluctuating system such as Ce or Yb. If the energy levels of more than two electronic configurations with different f-electron occupancy (n_f) are close each other, the quantum mechanical mixing tends to occur and valence fluctuation or mixing is generally observed. If the entropy becomes large because of the fact that the energy levels with different n_f are located close each other, it is likely that valence fluctuation also becomes significant. Actually, the behavior of the volume

magnetostriction at low temperatures in Fig. 2c is suggested to be in a similar physical situation of Ce changing its valence from nonmagnetic valence fluctuating (alpha) phase to magnetic integer valence (gamma) phase with elevating temperature. The numerical fitting procedure indicated a low Kondo temperature compare to the excitation energies ($E1^*$, $E2^*$). However, I think, in order to understand the physical situation better, more explanation is needed why the valence fluctuation effect is not significant although the different nf states locate close each other.

Response to (2): The limit $T_{fl} \ll E^*$ that we find for delta-Pu is the same as in gamma-Ce; the magnitudes of the energies also appear to be similar. We now discuss this more carefully in the final paragraph of the discussion. We speculate that this is the consequence of delta-Pu and gamma_Ce both being the largest volume phases (of Pu and Ce respectively) — larger volume being consistent with reduced wavefunction overlap. Both systems also appear to be close to integer valence in their largest volume phases, which is also accompanied by a small T_{fl} .

We have also changed two sentences in the first paragraph of the “magnetostriction” subsection so that an initial comparison is made with “materials with unstable f-electron shells” rather than to the narrower subcategory of valence fluctuating and Kondo lattice systems.

(3) Since the x-dependences of $E1^*$ and $E2^*$ are linear, it seems to be strange only 2% Ga doping suddenly stabilize the delta phase. Considering $F=E-TS$, where F is the free energy and E, T, S are internal energy, temperature, and entropy, respectively, the phase can be stabilized by increasing temperature at finite temperature. At $x=0$, delta-phase cannot be stable, because S is not large enough and an relaxation to the bottom of $E2$ curve (Fig3c) is large. With finite x, the three energy levels close each other and an enhancement S saves the energy. However, it is likely that x-dependence of $E1^*$ and $E2^*$ is small to realize the phase stability with relatively small x such as 2%-doping. I would like to ask authors give an explanation for this question.

Response to (3): The rate $\sim 100K$ per % at which E^*2 drops with increasing x is of comparable order to the rate at which the temperature of the phase transition (involving the largest enthalpy and volume change) drops with increasing x. We are not aware of any other energy scale that is found to change so rapidly with x in delta-Pu. However, as Reviewer #1 points out, both E and -TS must ultimately be considered (as well as any energy barriers) to determine whether a phase transition actually occurs. Rather than have an extended discussion in the main text, we have appended a new section to the Methods.

(4) In page 6, from line 4 to line7: Not until $T \geq 50$ K to the lattice constant data.

Does this sentence explain the behavior of curves shown in Fig 2e ? I cannot understand which graph is referred. Please revise it.

Response to (4): Fig. 2e is indeed being referred to; the sentence has now been reworked to make this clearer.

(5) I feel that Method part is too long compared to the main text. Outline of the fitting procedure can be included in the main text. It is because the fitting is essentially important to validate the conclusion of this paper. It would also be nice to include Table 1 in the main text. I think the fact that the magnetic moment of E2 (μ_{2^*}) is larger than that of (μ_{1^*}) explains the sign change of temperature variation of the volume magnetostriction. I also think that the obtained values except μ_{2^*} shown in Table 1 are worth explaining in the main text.

Responses to (5): Following the recommendations of Reviewer #1, the following changes have now been made: (i) The less critical parts of the Method have been put into a Supplementary Information document. (ii) An outline of the fitting procedure is included in Results section. (iii) Table 1 is included in the main text. (iv) A sentence referring to the large magnetic moments associated with E₂ is appended to the end of the 3rd paragraph of the Discussion section.

Reviewer #2 (Remarks to the Author):

In my view this manuscript cannot be accepted in its present form. I will explain my arguments below.

- The manuscript is written in an unclear style and is difficult to comprehend for someone only partially familiar with the techniques. For example figure 1a is put forward and cannot be understood without explanation, which is scattered in the manuscript. This should be corrected in my view.

Response: In response to recommendations of Reviewer #2, the introduction has been expanded to clarify the purpose of the manuscript and Fig. 1a has been moved to the Results section (it's now Fig. 3).

- The introduction contains information that is common knowledge and does not need to be included here. Figures 1b and 1c present no new data or evidence. They can be found in other works.

Response: Following the suggestions of Reviewer #2, Figs. 1b and c have now been removed.

- There is no information about the materials used for the measurements and their treatment. In this specific case this is important as Pu is a radioactive substance and the radioactive decay introduces changes in the materials. So information on the isotopic composition, age, thermal treatments are all required to assess the results of the manuscript.

Response: More information about the materials is now provided in the Methods.

- If I understand correctly, the phonon calculations have been made with a single value for the Debye temperature. This seems not correct to me, or at least I am missing the arguments why this can be done. The phonons are affected by the impurities (Ga, Al) and an change in the Debye temperature can be expected. Then also radiation plays a role. Lattice defects resulting from decay also affect the phonons, so the difference between the 2% and 7% Ga samples may be caused by other factors as well. In the approach employed here, everything is squeezed into an electronic contribution, whereas reality may be more complex.

Response: Following the above concerns of Reviewer #2, we now provide a justification for our neglect of the effect of Ga and Al concentration on the Debye temperature. In the last paragraph of the “Free energy modeling” subsection, we state that the Debye temperatures of the 2% and 7% Ga-substituted samples and the 5% Al-substituted samples are all very similar. We back this up in Supplementary Figure 3 by showing that the slopes of the C/T versus T^2 at low temperatures are very similar with each other and with the published data taken on 5% Al-substitution by Lashley et al. [PRL 91, 205901].

Furthermore, even if significant differences do occur in the phonon contributions, these ought not to impact the magnetostriction, which is sensitive almost exclusively to the electronic degrees of freedom. This is because the change in the volume of the sample is only ~ 2 ppm at $B=15T$, which makes any magnetic field effect on the phonons negligible.

- Figure 4 is not consistent with Figure 6. In the latter the 7% sample has a lower heat capacity than the 2% sample below 120 K, whereas in Figure 4d the difference is positive up to this temperature. I hope it is just a simple error in the presentation of both experimental and calculated data. Personally I doubt whether this difference is really significant, realising that the measurements were made on sample of 1 mg. This is especially true at high temperatures, where the contribution from the puck become big compared to that of the sample. The authors claim that the difference is not affected by the subtraction of the addendum, but I am not convinced by this argument. Only if the same addendum curve can be applied, but then the same puck should be used. But the self-heating effect is slight different, and I guess the amount of grease to fix the samples too. Therefore, the error of about 2 J K⁻¹ mol⁻¹ seems underestimated in my view.

Response: We thank Reviewer #2 for pointing out a mislabeling of the vertical axis in Fig. 4d; the order of the 2% and 7% subscripts had been reversed. We thank the Reviewer also for raising concerns regarding the approximate mass of the sample in relation to the accuracy of heat capacity measurements. The actual masses of the samples used in heat capacity measurements (5.5mg and 4.1mg) are now included in the Methods, as are more details concerning the grease and the use of the same addendum throughout.

Overall, interesting paper, interesting results, but in the end the conclusion is the result of fitting of a model, where for my feeling the neglect of other effects might lead to a overestimation of the electronic entropy.

Reviewer #3 (Remarks to the Author):

I read with high interest this paper on magnetostriction measurements in plutonium.

As a theoretician, I cannot judge some technical part of the paper.

It seems however that measurements are precise and thus useful and represent an important contribution to the physics of plutonium.

I have three main comments :

1. The model used to explain the non monotonous magnetostriction effect is interesting.

Indeed the description of temperature effect is very important.

Low temperature description of actinides can be obtained by various method, see Ref 13 and PRX 5 011008 (2015), and JPCM 30 405603 (2018) for recent works. These works present structural properties of phases of plutonium at 0 K.

These papers are in nice agreement with the thermodynamics at 0 K of phases of plutonium in the paper (PRB 58, 15433 (1998)).

So 0K properties are described by theories.

Moreover in this last paper by Wallace, it is underlined that electronic entropy effect and anharmonic effect cannot be disentangled.

This is also in agreement with the paper of Manley et al on phonons which reports an excess entropy. Would it be possible that this excess entropy comes from anharmonic phonon effect instead of electronic effect as proposed in the manuscript ? It is a key point that should be addressed in the current paper.

Response to 1: Reviewer #3 raises an important point regarding the electronic and anharmonic phonon contributions to the thermodynamics, which had previously proved challenging to separate. In view this problem, Wallace combined them into a single T^2 contribution to the free energy. Meanwhile Manley et al estimated the magnitude of the anharmonic contribution and determined that it is too small for it alone to explain the delta to alpha transformation. Both Manley et al and Jeffries et al refer to a “missing entropy” in this context.

Magnetostriction measurements provide a means to resolve this situation because they are almost exclusively sensitive to the electronic degrees of freedom (they are insensitive to phonons, except via higher order effects: for example, because magnetostriction changes the size of the sample, phonons can still in principle contribute to the magnetostriction. However, the change in the volume of the sample is so small [only ~2ppm at B=15T] that the magnetic field effect on the phonons is negligible).

Following Reviewer #3's question, we have now slightly expanded the discussion of the missing entropy in the introduction. Furthermore, we now state the important points concerning the ability of magnetostriction measurements to isolate the electronic

contribution in the second paragraph of the introduction. We have also made sure that all of the above references are cited.

2. The comparison to Cerium is surprising. In cerium, there is a local moment in the gamma phase and it explains the important role of the magnetic field and the entropy (see the cited paper of Drymiotis et al).

In delta plutonium, there is no magnetic moment (cf Ref 14). So it is basically a different physics. In cerium, it seems to me that the current explanation of entropy is just a fluctuation of the electron in the f orbitals and not the valence fluctuation.

It should be commented in the paper.

Response to 2: Reviewer #3 raises concerns over the suitability of a comparison of Pu with Ce. We have now revised the final paragraphs of the discussion to clarify the similarities and differences between delta-Pu and gamma-Ce. The similarity is that both delta-Pu and gamma-Ce appear to be in a limit where the valence fluctuation temperature scale is significantly less than the excitation energies. This is an important factor in causing a volume instability to occur in delta-Pu as well as in gamma-Ce and YbInCu4.

However, it's also important to state the differences between delta-Pu and gamma-Ce. Whereas gamma-Ce has a magnetic lowest energy configuration, delta-Pu appears to have a non-magnetic lowest energy configuration and multiple excited configurations. This distinction is now stated explicitly in the manuscript.

3. A recent work JPCM 29, 245402 (2017) discuss the role of impurity on the dynamical stability of delta plutonium.

Are the conclusion of this paper in agreement with the vision proposed in the manuscript?

Response to 3: The primary result of the above JPCM manuscript is that delta-Pu is dynamically unstable with regards to phonons, but this is no longer the case when Ga or Al is substituted. This finding is consistent with experimental observations, yet the impact on the free energy is unclear. The degree to which the phonon dispersion turns negative is ~5 meV, suggesting that it involves smaller energy perturbation than E^*_1 and E^*_2 .

In the Methods, we now cite this paper and append the sentence "Dynamical instability of the delta-phase lattice in pure Pu is another possible factor, although the energy scale appears to be lower than that of the electronic excitations" to the end of the first paragraph of the Discussion section.

In view of the question, it really seems that the paper is interesting, but I need clarification on these questions.

Moreover, the writing is somewhat unclear, in particular the conclusion could be rewritten to be more clear. In particular, do the virtual fluctuations are important or not ? Using shorter sentences, and clear ideas would be helpful for the reader.

Response: A statement pertaining to the relevance of valence fluctuations is now made at the end of the introduction, with a longer discussion about valence fluctuations being given at the end of the Discussion section.

REVIEWERS' COMMENTS:

Reviewer #1 (Remarks to the Author):

The structure of the manuscript has been much improved and the statements that were not very clear for me have also been revised adequately. I feel like that having local multi-configuration-states can be a common nature of 5f electron system considering balance between degrees of itinerancy and localization, and so findings in this work generally contribute to research of solid states. I agree with publication of this manuscript.

Reviewer #2 (Remarks to the Author):

I am happy with the response of the authors. The manuscript is now easier to read and understand. It does not take away my concern that the neglect of dilution effects in the model might lead to a overestimation of the electronic entropy, but the arguments put forward in the article are clear and justified, so In my view it can now be published.

Reviewer #3 (Remarks to the Author):

I read the new version of the manuscript.
I think the paper is more readable now concerning the point that I raised. It is however sometime difficult to see the complete picture of the paper.
However, the experimental results seems really interesting and should raise questions and some theoretical works.

A useful clarification:

The authors should carefully explain in the introduction the difference between "virtual valence fluctuations" and "f-electron shell instabilities" to increase the readability of the paper.

See response to Reviewer 3 below:

Reviewer #1 (Remarks to the Author):

The structure of the manuscript has been much improved and the statements that were not very clear for me have also been revised adequately. I feel like that having local multi-configuration- states can be a common nature of 5f electron system considering balance between degrees of itinerancy and localization, and so findings in this work generally contribute to research of solid states. I agree with publication of this manuscript.

Reviewer #2 (Remarks to the Author):

I am happy with the response of the authors. The manuscript is now easier to read and understand. It does not take away my concern that the neglect of dilution effects in the model might lead to an overestimation of the electronic entropy, but the arguments put forward in the article are clear and justified, so in my view it can now be published.

Reviewer #3 (Remarks to the Author):

I read the new version of the manuscript.
I think the paper is more readable now concerning the point that I raised. It is however sometimes difficult to see the complete picture of the paper.
However, the experimental results seem really interesting and should raise questions and some theoretical works.

A useful clarification:

The authors should carefully explain in the introduction the difference between "virtual valence fluctuations" and "f-electron shell instabilities" to increase the readability of the paper.

Response from authors: Following the suggestion of Reviewer #3, definitions of "virtual valence fluctuations" and "f-electron shell instabilities" have been included in the introduction for clarification purposes.